# IK1 channels do not contribute to the slow afterhyperpolarization in pyramidal neurons

**Kang Wang[1†], Pedro Mateos-Aparicio[2†], Christoph Hönigsperger[2], Vijeta Raghuram[1], Wendy W Wu[3‡], Margreet C Ridder[4], Pankaj Sah[4], Jim Maylie[3], Johan F Storm[2], John P Adelman[1*]**

[1]Vollum Institute, Oregon Health and Science University, Portland, United States; [2]Department of Physiology, Institute of Basic Medical Sciences, University of Oslo, Oslo, Norway; [3]Department of Obstetrics and Gynecology, Oregon Health and Science University, Portland, United States; [4]Queensland Brain Institute, The University of Queensland, Brisbane, Australia

**Abstract** In pyramidal neurons such as hippocampal area CA1 and basolateral amygdala, a slow afterhyperpolarization (sAHP) follows a burst of action potentials, which is a powerful regulator of neuronal excitability. The sAHP amplitude increases with aging and may underlie age related memory decline. The sAHP is due to a $Ca^{2+}$-dependent, voltage-independent $K^+$ conductance, the molecular identity of which has remained elusive until a recent report suggested the $Ca^{2+}$-activated $K^+$ channel, IK1 (KCNN4) as the sAHP channel in CA1 pyramidal neurons. The signature pharmacology of IK1, blockade by TRAM-34, was reported for the sAHP and underlying current. We have examined the sAHP and find no evidence that TRAM-34 affects either the current underling the sAHP or excitability of CA1 or basolateral amygdala pyramidal neurons. In addition, CA1 pyramidal neurons from IK1 null mice exhibit a characteristic sAHP current. Our results indicate that IK1 channels do not mediate the sAHP in pyramidal neurons.

*For correspondence: adelman@ohsu.edu

†These authors contributed equally to this work

Present address: ‡U.S. Food and Drug Administration, Silver Spring, United States

Competing interests: The authors declare that no competing interests exist.

## Introduction

In 1980, Hotson and Prince described a long lasting hyperpolarization (AHP) that followed current-induced repetitive firing of action potentials in CA1 pyramidal neurons (*Hotson and Prince, 1980*). Since this initial finding the role of the slow afterhyperpolarization (sAHP) has been elucidated particularly in the hippocampus where its activity profoundly impacts learning. Thus, for example, the sAHP increases with aging and ovarian hormone deficiency (*Wu et al., 2011*) and this may underlie cognitive deficits in older or hormone deficient individuals. In 1982, Madison and Nicoll demonstrated that this sAHP was blocked by application of noradrenaline, via protein kinase A (PKA), since the effect is prevented by PKA antagonists and mimicked by application of cyclic AMP or the PKA catalytic subunit (*Madison and Nicoll, 1982*; *Pedarzani and Storm, 1993*). Blocking the sAHP eliminated spike-frequency adaptation and resulted in dramatically enhanced repetitive firing (*Madison and Nicoll, 1982*). This same effect can be observed following application of other neurotransmitter receptors, such as histamine (*Haas and Konnerth, 1983*), dopamine (*Benardo and Prince, 1982a*; *Pedarzani and Storm, 1995*), cholinergic agonists (*Benardo and Prince, 1982b*; *Benardo and Prince, 1982c*; *Cole and Nicoll, 1984*), and various peptides (*Haug and Storm, 2000*). In 1984, these same authors reported that loading cells with the $Ca^{2+}$ chelator, EGTA, or extracellular application of the voltage-gated $Ca^{2+}$ channel blocker, $Cd^{2+}$, blocked the sAHP (*Madison and Nicoll, 1984*). In 1986, Lancaster and Adams used a hybrid clamp technique to record

**eLife digest** Neurons carry signals in the form of electrical impulses called action potentials. These nerve impulses result from ions flowing through proteins called ion channels in the neuron's membrane, and they determine how the neuron communicates with neighboring neurons. The number of action potentials a neuron can produce can vary over a wide range. In the brain, a particular kind of ion channel limits the number of action potentials that many neurons produce via a negative feedback mechanism. That is to say, nerve impulses activate this ion channel and the activated channel then makes the neuron less able to send further nerve impulses for a while. The activity of this ion channel increases with age and it may be responsible for some forms of age-related decline in cognitive abilities. However, the exact identity of the ion channel responsible was unclear.

Recent research has suggested the ion channel in question was a protein called IK1. This conclusion was largely based on how this ion channel responded to drugs in the laboratory. Wang, Materos-Aparico et al. sought to verify this conclusion and, in contrast with the previous reports, found that the IK1 ion channel did not respond to these drugs in the same way when it was in neurons in the brains of mice.

In further experiments, mice that had been engineered to lack the IK1 ion channel still showed the characteristic negative feedback that regulates the firing of action potentials. Thus, Wang, Materos-Aparico et al. found no evidence to support the previous conclusion, and instead conclude that the exact identity of this important ion channel in the brain has yet to be defined.

the sAHP in current clamp and the underlying current in voltage clamp. This showed a characteristic slowly activating current, the IsAHP, with a reversal potential that shifted with the concentration of extracellular $K^+$ in a manner consistent with a $K^+$ current (*Lancaster and Adams, 1986*). In 2002, Power et al. showed that activation of the IsAHP was voltage independent (*Power et al., 2002*). Taken together, these and other reports strongly support the proposal that the sAHP was due to the activity of a $Ca^{2+}$-dependent, voltage-independent $K^+$ channel. Given the importance of the sAHP the identity of the underlying channel(s) and associated molecular components are of great interest. In 1996, Kohler et al. reported the cloning and expression of three $Ca^{2+}$-activated, voltage independent $K^+$ channels, the SK (small conductance) channels (SK1-3; KCNN1-3) (*Kohler et al., 1996*) and in 1997 the fourth member of this family, SK4 (IK1; KCNN4; intermediate conductance) was reported (*Ishii et al., 1997*; *Joiner et al., 1997*). The pharmacological fingerprints of these channels are distinct. Apamin, an 18 amino acid peptide isolated from honeybee venom selectively blocks SK2 and SK3 channels, while SK1 is less apamin sensitive and IK1 is not apamin sensitive (*Adelman et al., 2012*). Importantly, the sAHP is not blocked by apamin. IK1 (SK4) is blocked by charybdotoxin (ChTX) a 37 amino acid peptide isolated from scorpion venom that also blocks BK channels (*Ishii et al., 1997*). The antimycotic agent, clotrimazole also blocks IK1 (*Ishii et al., 1997*), but this is also a P450 inhibitor. However, the related triarlymethane, TRAM-34, potently and specifically blocks IK1 channels (*Wulff et al., 2000*). Sensitivity to TRAM-34 has therefore been taken as the signature of IK1 channels.

The biophysical and pharmacological properties of the SK/IK channels fulfill many of the expected characteristics of the channels underlying the sAHP in CA1 and basolateral amygdala (BLA) pyramidal neurons: $Ca^{2+}$-activated, voltage-independent, $K^+$ selective. *In situ* hybridization and immunohistochemistry suggest that SK1-3 are widely expressed in overlapping yet distinct patterns in the brain, including hippocampus and amygdala (*Stocker and Pedarzani, 2000*; *Sailer et al., 2002*). IK1 mRNA expression was detected in some limited brain areas but hippocampus and amygdala were ambiguous (http://mouse.brain-map.org/experiment/show/130911; http://mouse.brain-map.org/experiment/show/119686). Knock out mice for each of the three SK channels revealed that the apamin-insensitive IsAHP was not affected in any of the SK null mice, eliminating them as candidates for the sAHP channel (*Bond, 2004*). However, a recent report used an IK1 reporter mouse to show that the IK1 promoter was active in several brain regions including hippocampus (*Turner et al., 2015*). Turner and colleagues further reported that the sAHP and the IsAHP in CA1 pyramidal neurons were greatly reduced by application of TRAM-34, and the sAHP was absent in CA1 pyramidal neurons

from IK1 null mice, strongly suggesting that IK1 channels underlie the sAHP (*King et al., 2015*). In line with this, a recent paper reported that IK1 channels were suppressed by direct phosphorylation by PKA (*Wong and Schlichter, 2014*). Given the disparity on the role of IK1 channels, we have examined the sensitivity of the sAHP in pyramidal neurons from area CA1 of the hippocampus and the BLA to TRAM-34 and found that this compound did not significantly affect the current underlying the sAHP measured in voltage clamp. TRAM-34 also had no effect on the sAHP amplitude or intrinsic excitability measured in current clamp. Moreover, IK1 null mice express a characteristic IsAHP. Together our results indicate that IK1 channels do not mediate the sAHP in pyramidal neurons.

## Results

### The IsAHP in CA1 hippocampal pyramidal neurons is not affected by TRAM-34

As previously described (*Pedarzani and Storm, 1993*; *Madison et al., 1987*; *Gerlach et al., 2004*), a robust IsAHP was recorded in whole-cell voltage clamp configuration (22–24°C) from CA1 pyramidal neurons in freshly prepared hippocampal slices from 6–8 week old rats. From a holding potential of -63 mV, a 200 ms voltage command to +7 mV was delivered to promote $Ca^{2+}$ influx through voltage-gated $Ca^{2+}$ channels. Repolarization to -63 mV elicited a characteristic slowly decaying outward tail current, the IsAHP that decayed over several seconds with a time constant of $2.9 \pm 0.2$ s (n = 20). Apamin was included in the bath solution to eliminate the SK channel contribution that overlaps with the initial decay phase of the IsAHP (*Bond, 2004*). The IsAHP was measured as the current at 1 sec after the voltage step. The tail current protocol was repeated every 30 sec for 25 min, and showed modest rundown of the IsAHP in control cells, being reduced to $0.74 \pm 0.17$ of the initial current amplitude (n = 12, P < 0.001). In some experiments carbachol (CCh; 1 µM), a muscarinic agonist that potently blocks the IsAHP, was applied after 25 min (*Figure 1B-D*). To test the effects of TRAM-34, control tail currents were first obtained for 5 min in the absence of drug. The average amplitude of the IsAHP in this baseline control period was $209.3 \pm 27.9$ pA (n = 11), not different for control cells ($195.0 \pm 28.3$ pA, n = 12) (*Figure 1C,D*). TRAM-34 (1 µM) was added to the bath solution and the tail current protocol was continued. After 25 min in TRAM-34, the IsAHP relative to control baseline was $0.79 \pm 0.18$ (n = 11), not different than rundown in control cells (*Figure 1E*). As in control, subsequent addition of CCh abolished the IsAHP (*Figure 1C*). While TRAM-34 rapidly blocks native and cloned IK1 channels when applied in the bath solution, the binding site for TRAM-34 is internal (*Wulff, 2001*). Therefore, TRAM-34 was also applied through the patch pipette (*Figure 1A,D-F*). With intracellular dialysis of TRAM-34, the relative amplitude of the IsAHP measured 25 min after dialysis was not different from external TRAM-34 application ($0.92 \pm 0.13$ of the initial current, n = 4); CCh treatment eliminated the IsAHP. The sensitivity of the IsAHP to 5 µM TRAM-34 was also tested and this increased concentration of TRAM-34 was without effect ($0.83 \pm 0.07$, n = 9). ChTX (100 nM) was also tested and as for TRAM-34, ChTX did not affect the IsAHP (relative IsAHP $0.83 \pm 0.11$, n = 10). To be certain that the drugs, TRAM-34 and ChTX, were active each was bath-applied to HEK293 cells transiently expressing cloned IK1 channels. TRAM-34 (1 µM) produced a rapid block within 30 sec of the IK1 current (relative current after TRAM-34 compared to baseline = $0.07 \pm 0.03$, n = 3) (*Figure 2*). Similarly, ChTX (100 nM) blocked cloned IK1 currents (relative current after ChTX = $0.05 \pm 0.01$, n = 8; not shown). These data show that ChTX or TRAM-34 does not block the IsAHP but they do block IK1 channels.

### The sAHP and excitability of CA1 pyramidal neurons are not affected by TRAM-34

Somatic whole-cell current clamp recordings (33°C) were obtained from CA1 pyramidal neurons in rat hippocampal slices. A brief spike train was evoked by depolarizing current injection (200 pA for 100 ms), and was followed by characteristic medium (m) and slow (s) AHPs (*Figure 3A*). The peak AHP amplitudes recorded in normal aCSF were $4.36 \pm 0.39$ mV for the mAHP, and $2.79 \pm 0.41$ mV for the sAHP (n = 6). The cells treated with TRAM-34 for 30 min (n = 7) showed similar AHP amplitudes: $4.46 \pm 0.33$ mV for the mAHP and $3.17 \pm 0.27$ mV for the sAHP (*Figure 3B*). In a different set of neurons (n = 5) TRAM-34 application for at least 25 min did not affect the sAHP (*Figure 3D*), but

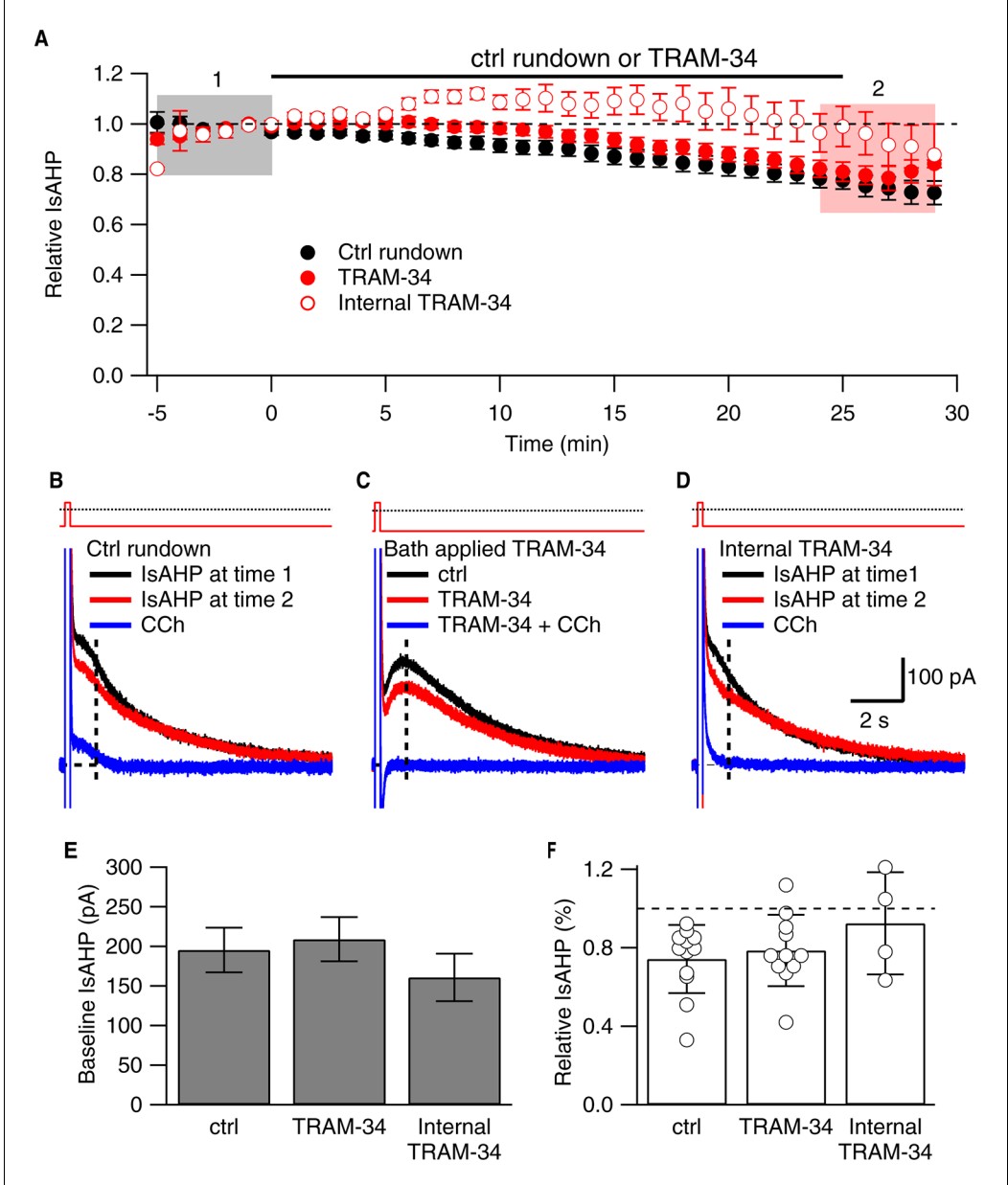

**Figure 1.** TRAM-34 (1 µM; 22-24°C) does not affect the IsAHP. (**A**) Time course of the normalized amplitude of the IsAHP from control rundown (ctrl, closed black symbols), or TRAM-34 treated cells either bath applied (closed red symbols) or internally delivered (open red symbols) in CA1 pyramidal neurons. (**B, C, D**) Representative CA1 pyramidal neuron tail currents elicited by the voltage protocol shown above the traces for control rundown (**B**), bath applied TRAM-34 (**C**) and internally delivered TRAM-34 (**D**) at −5 to 0 min (black), 25-30 min (red) and 10– 15 min after CCh application (blue). Vertical dash line at 1 sec after the pulse indicates time point for IsAHP measurement in time course plot of (**A**). (**E**) Bar plot of the amplitudes of the IsAHP during a 5 min baseline period for control rundown (Ctrl), bath applied TRAM-34 and internal TRAM-34. (**F**) Bar plot of the IsAHP measured at 25–30' (red shaded area panel A) relative to 5 min baseline (black shaded area panel A) for control rundown (Ctrl; n = 10) and bath applied TRAM-34 (n = 11) and internal TRAM-34 (n = 4). Error bars are ± SEM.

the sAHP was rapidly blocked by subsequent bath application of noradrenaline (10 µM) (*Figure 3C, D,E*). Combined bath application of XE991 to block Kv7/KCNQ/M channels and apamin to block SK channels, a combination also used by *King et al. (2015),* for at least 30 min effectively eliminated the mAHP, but did not significantly affect the sAHP (*Figure 4A,B*). The same combination including TRAM-34 also did not affect the sAHP (*Figure 4A,B*). Similar results were obtained either using acute hippocampal slices or hippocampal slice cultures, so the data were pooled together (*Figure 5*).

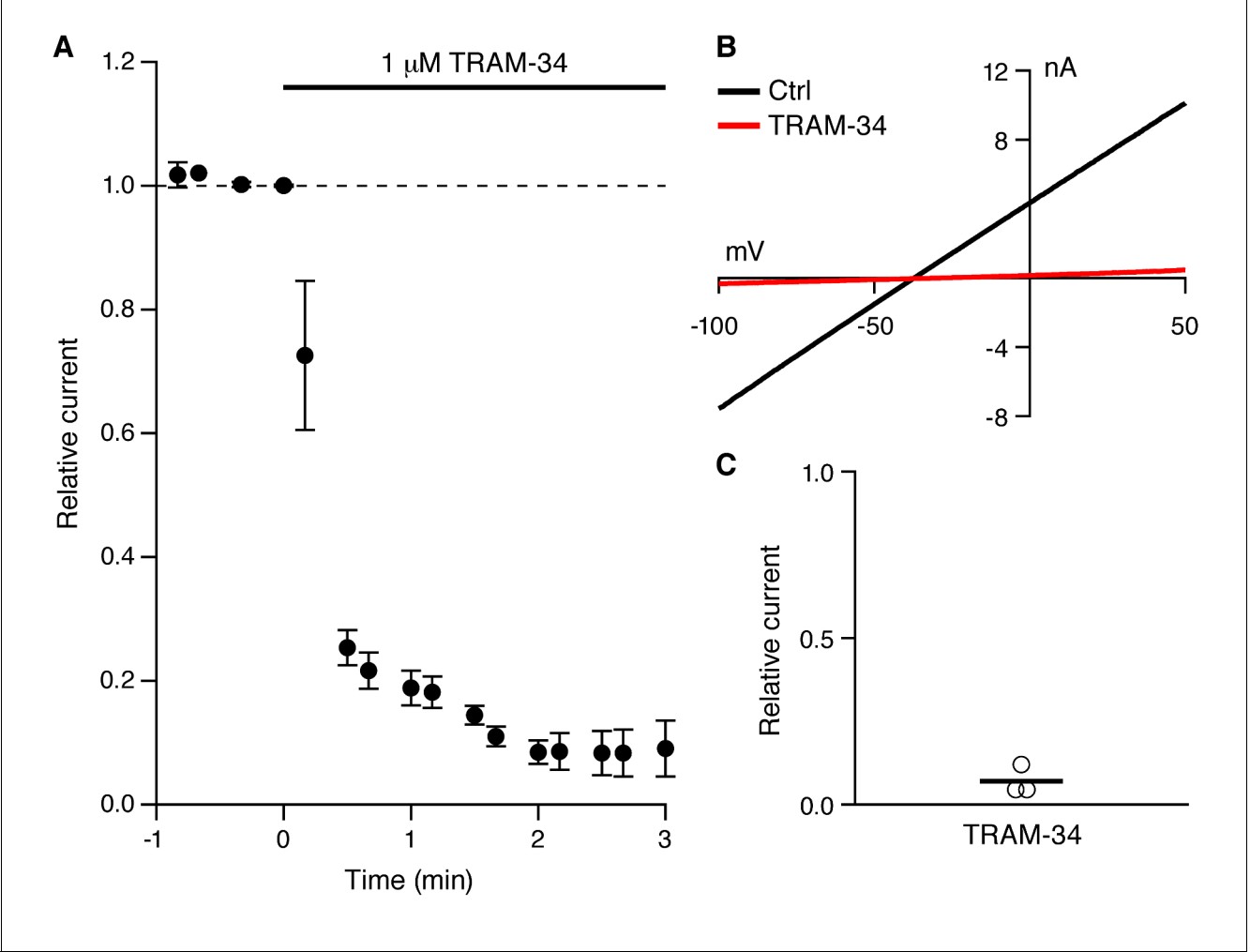

**Figure 2.** TRAM-34 blocks cloned IK1 channels. (**A**) Time course of TRAM-34 block of IK1 channels expressed in HEK293 cells (n = 3). (**B**) Representative whole-cell recordings with 10 µM Ca$^{2+}$ in the patch pipette. Currents were evoked from HEK293 cells expressing IK1 by voltage ramp commands (0.16 mV/ms) in control bath solution (black) and after TRAM-34 application (red) (1 µM; 22– 24 ℃). (**C**) Scatter plot of TRAM-34 block of IK1 current (n = 3).

Finally, intrinsic excitability was examined in CA1 pyramidal cells in acute hippocampal slices. Spike trains were evoked by a series of depolarizing current pulses (0-400 pA; 1s) in control or TRAM-34 containing bath solution (*Figure 5A*). There was no difference in spike rates between the two groups of cells (*Figure 5B*). These data indicate that TRAM-34 does not affect the sAHP or intrinsic excitability.

## The IsAHP, the sAHP, and excitability of pyramidal neurons in amygdala are not affected by TRAM-34

Pyramidal neurons of the basolateral amygdala (BLA) have been shown to express an IsAHP and sAHP that are indistinguishable from those observed in hippocampal CA1 pyramidal neurons (*Power et al., 2011*), suggesting that the same molecular components underlie the sAHP. BLA pyramidal neurons were first recorded in whole-cell voltage clamp (*Figure 6A,B*). From a holding potential of −50 mV a depolarizing command to 10 mV was given for 100 ms. Upon return to −50 mV a characteristic outward tail current, IsAHP, was observed (*Figure 6A*). Neurons were recorded in the absence (n = 8) or presence (n = 10) of TRAM-34 (1 µM) in the internal pipette solution. The IsAHP showed modest rundown when examined at 2, 9 and 17 min after whole-cell formation but TRAM-34 was without effect (*Figure 6B*). In either condition, subsequent addition of noradrenaline (10 µM)

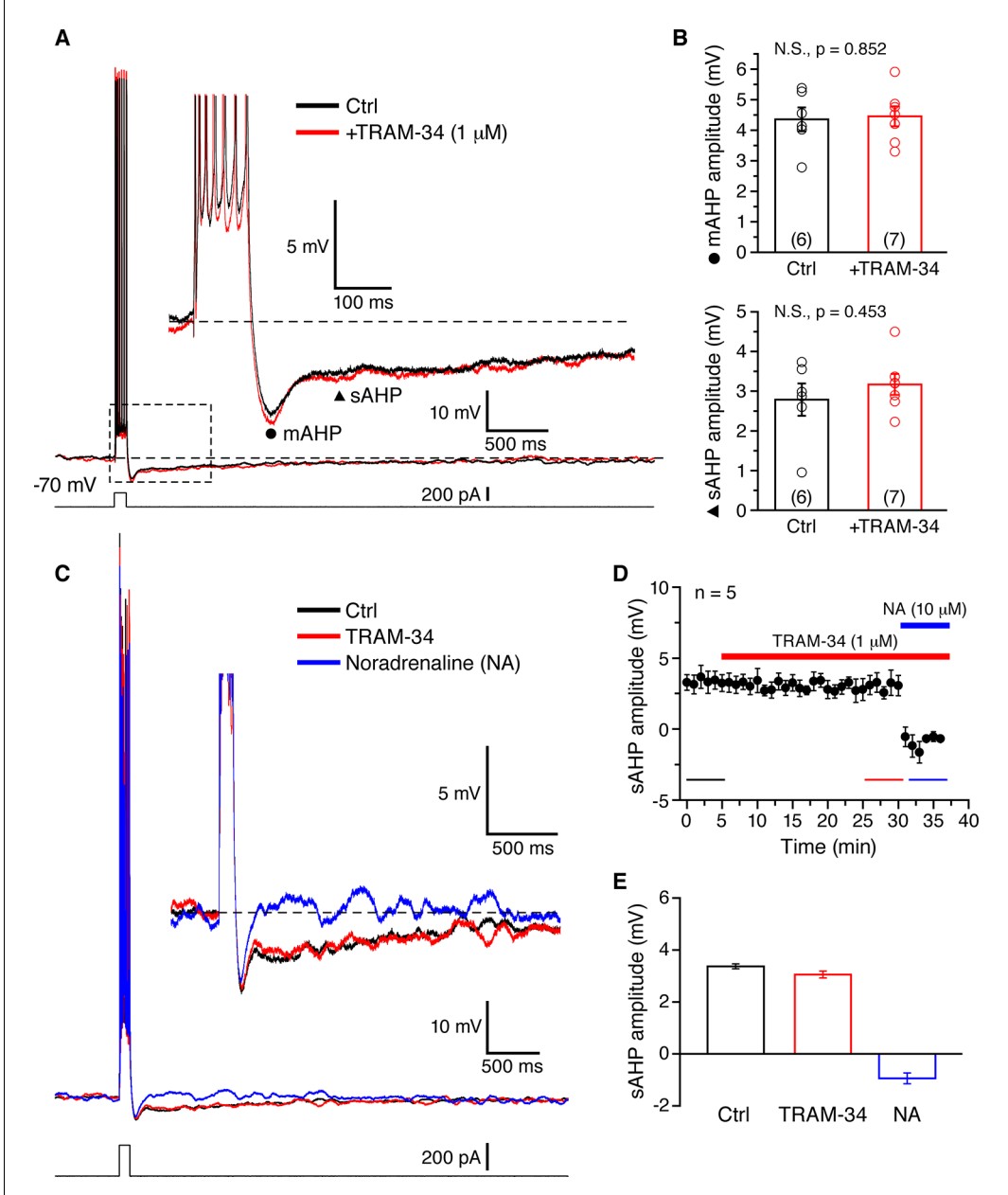

**Figure 3.** TRAM-34 has no significant effects on the medium afterhyperpolarization (mAHP) or slow afterhyperpolarization (sAHP) in CA1 pyramidal cells. (A) Brief spike trains (7 spikes/100 ms) were evoked by depolarizing current pulses from a holding potential of -70 mV. Black and red traces represent recordings from two pyramidal cells, in control medium and after incubation with TRAM-34 (1µM) for 30 minutes. Inset, showing mAHP(●) and sAHP (▲) at an enlarged scale. (B) mAHP and sAHP amplitudes were not significantly different between control and TRAM-34 treated groups. Data are given as, mean ± SEM. (C) Representative traces of the effect of TRAM-34 (1 µM, red) and noradrenaline (NA, 10 µM, blue) on the mAHP and sAHP in CA1 pyramidal neurons. (D) Time course of the sAHP amplitude measured in panel A. Bath application of TRAM-34 (1 µM) for 25 min did not significantly reduce the sAHP (n = 5). However, subsequent application of noradrenaline (NA, blue line) rapidly eliminated the sAHP. (E) Summary bar plots showing the sAHP amplitude averaged over a period of five minutes during control (black bar), TRAM-34 (red bar) and NA (blue bar) application. The averaged periods are represented by color bottom lines in panel D.

abolished the IsAHP (n = 1 in the absence and n = 2 in the presence of TRAM-34). BLA pyramidal

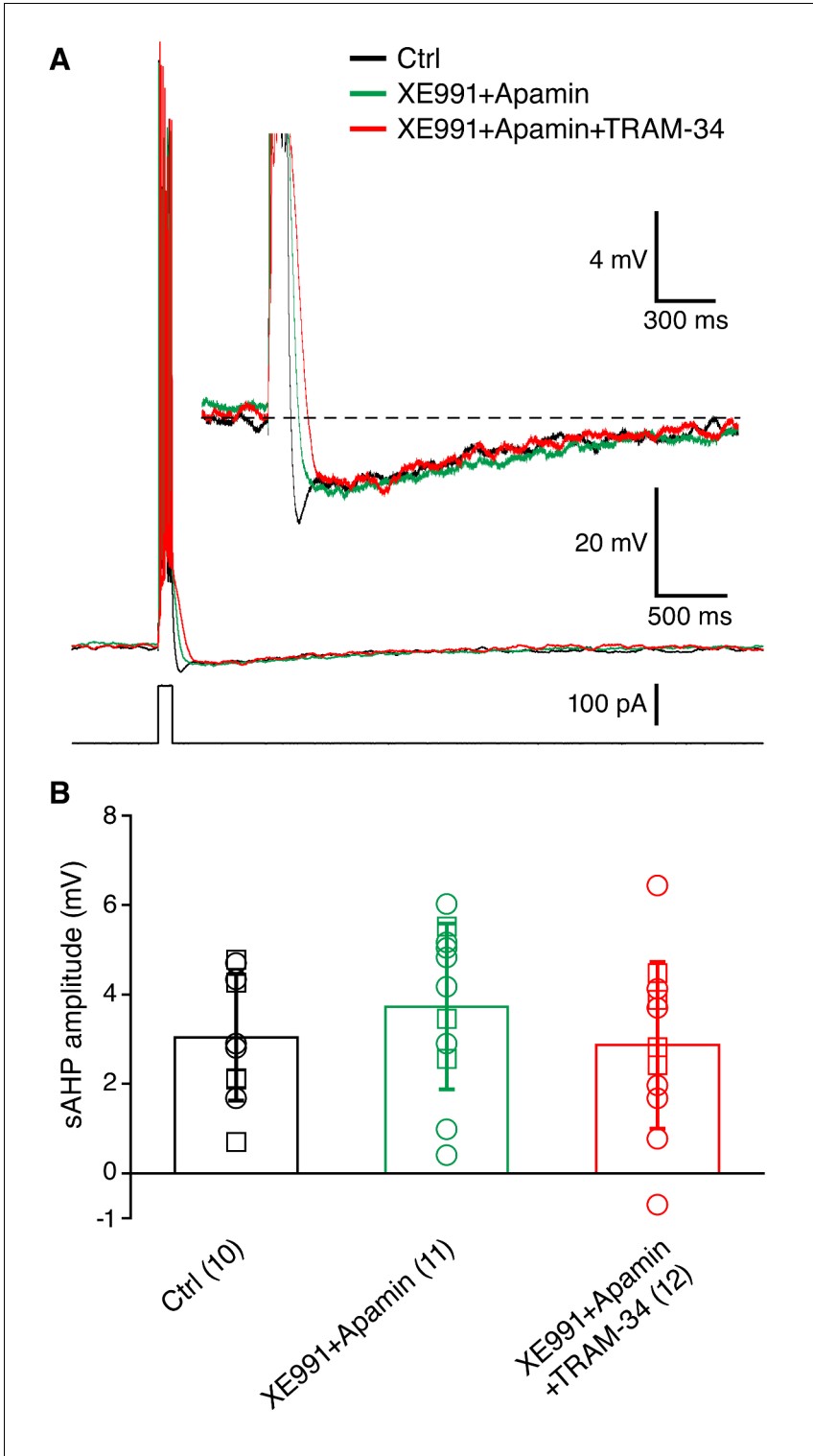

**Figure 4.** Incubation of acute hippocampal slices and organotypic slice cultures with TRAM-34 (1 μM) did not reduce the sAHP in CA1 pyramidal neurons. (**A**) Example traces showing the mAHP and sAHP in three different conditions tested. Cells in the control group (black trace) were recorded in normal ACSF. Cells in the second group (XE991+Apamin, green trace) were recorded after incubation for at least 30 minutes in ACSF with XE991 (10 μM) and apamin (100 nM). Cells in the third group (XE991+Apamin+TRAM-34, red trace) were recorded after incubation for at least 30 minutes in ACSF with XE991 (10 μM), apamin (100 nM) and TRAM-34 (1 μM). (**B**) Summary of the results from slices (open circles) and organotypic cultures (open squares). The sAHP amplitude did not differ between the three groups of cells: (1) control (n = 10), (2) XE991+Apamin (n = 11), and (3) XE991+Apamin +TRAM-34 (n = 12).

neurons were also recorded in current clamp mode (*Figure 6C,D*). A train of action potentials was evoked every 10 sec by an 800 ms depolarizing current injection. Whether recorded in the absence (n = 6) or presence (n = 7) of TRAM-34 in the internal solution the numbers of action potentials at 1 min or 18 min were not significantly different (*Figure 6D*, top). In addition, TRAM-34 did not affect action potential half width (*Figure 6D*, bottom) or resting membrane potential (not shown). These results show that TRAM-34 does not affect the IsAHP or excitability in BLA pyramidal neurons.

### IK1 null mice express the IsAHP

Whole-cell voltage clamp recordings were made from CA1 pyramidal neurons in freshly prepared hippocampal slices from IK1 null mice (*Si, 2006*) or strain-matched wild type mice, as for those in rat (above). In wild type mice, the amplitude of the slow component of the outward tail current measured at 1 s following repolarization to − 63 mV was 78.2 ± 16.5 pA (n = 9), and was blocked by subsequent application of CCh (*Figure 7A,C*). The IsAHP elicited from CA1 pyramidal neurons of IK1 null mice was 74.9 ± 13.8 pA (n = 12) and was potently blocked by CCh. Current subtraction yielded the CCh-sensitive IsAHP current with the characteristic slow rising onset and slow decay (*Figure 7B, C*). Thus, CA1 pyramidal neurons from IK1 null mice express an IsAHP that seems indistinguishable from that of wild type mice.

## Discussion

The sAHP conductance in CA1 pyramidal neurons is a powerful modulator of intrinsic excitability and many neurotransmitters activate second messenger pathways that converge on the sAHP, suppressing the sAHP and increasing intrinsic excitability (*Haug and Storm, 2000*). Modulation of the sAHP has been implicated in behavioral learning: animals with a smaller sAHP in CA1 pyramidal neurons learn hippocampus-dependent tasks better than those with a larger sAHP (*Moyer et al., 2000*; *Tombaugh, 2005*), and a reduction of the sAHP is observed after successful learning (*Moyer et al., 1996*; *Oh, 2003*). The sAHP increases with age (*Landfield and Pitler, 1984*), and this may underlie cognitive deficits in older animals (*Deyo et al., 1989*; *Knuttinen et al., 2001*). Thus, understanding the molecular basis of the sAHP is important and may lead to novel therapeutic approaches to manage cognitive decline. Compelling evidence suggests that the sAHP reflects the activity of $Ca^{2+}$-

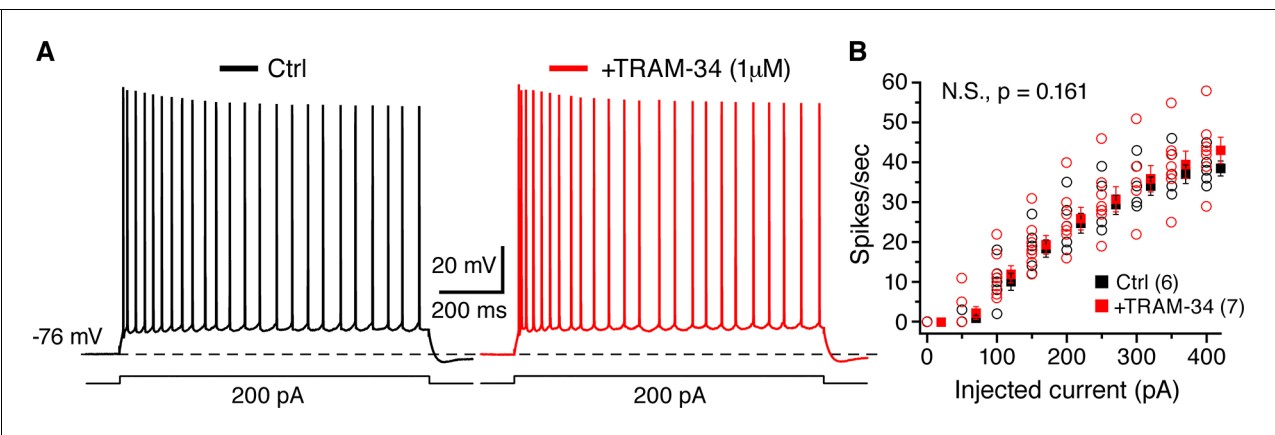

**Figure 5.** TRAM-34 (1 µM) had no significant effect on the excitability of CA1 pyramidal cells. (**A**) Representative spike trains evoked by 1s long depolarizing current (200 pA) injections from -76 mV, recorded from pyramidal cells in control medium (left, black trace) and after incubation of TRAM-34 (right, red trace). (**B**) Comparison of spike rates (spikes/s) between control (black) and TRAM-34 (red) treated groups evoked by depolarizing, 1 s long current pulses (0–400 pA). Mono-exponential fits were used to compare spike rates in control medium and TRAM-34 treated groups, by using the function: $f(X) = A[exp(-x/\tau)] + y0$. No significant differences were found between control and TRAM-34 treated groups [control, τ: 263 (58) pA; TRAM-34, τ: 371 (65) pA, N.S, p = 0.161, *t*-test after Box-Cox transformation (Minitab 17)].

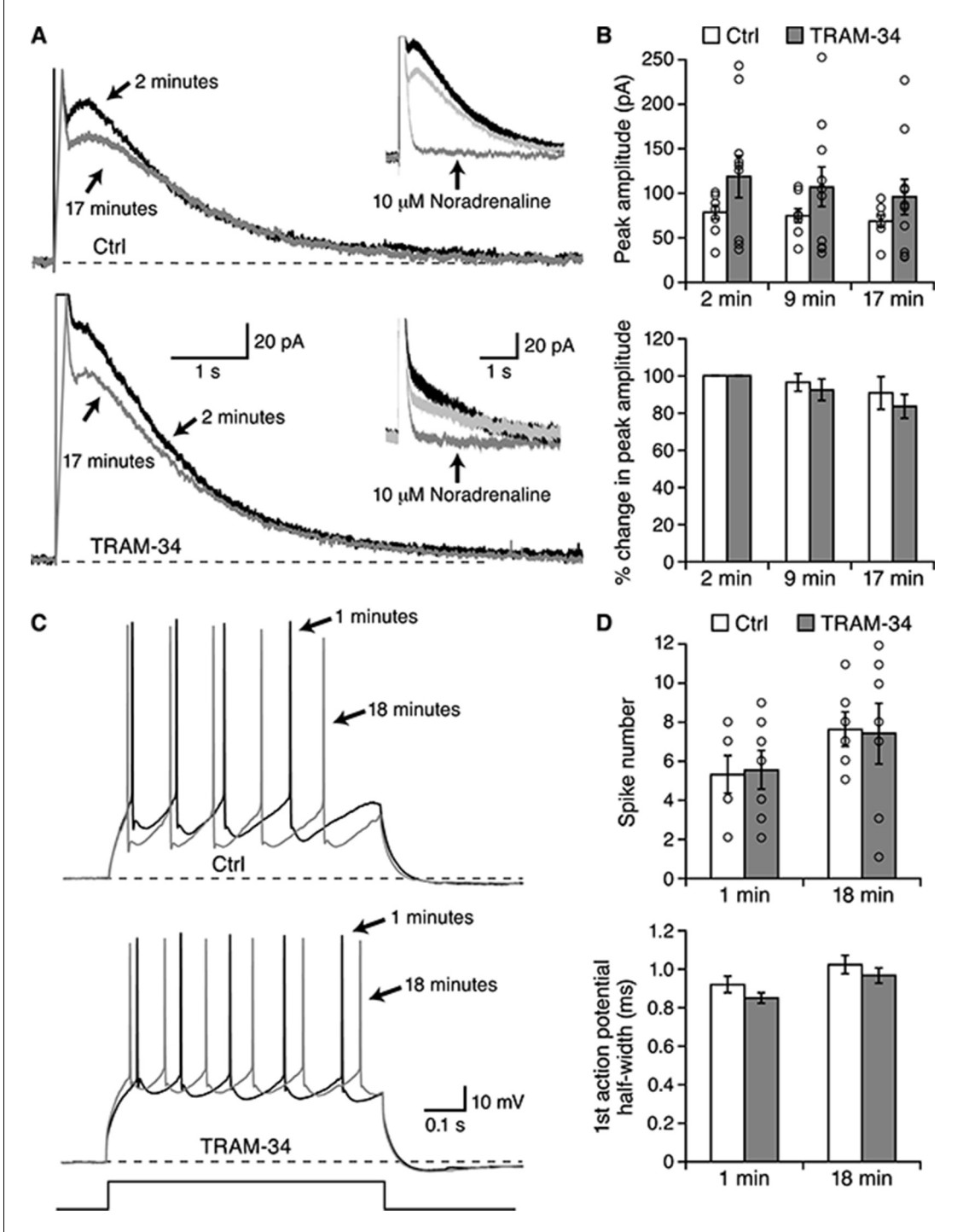

**Figure 6.** TRAM-34 did not block the IsAHP in BLA pyramidal neurons. Whole-cell voltage clamp recording from BLA pyramidal neurons. (**A**) Representative current traces for IsAHP current evoked from a holding potential of −50 mV under control conditions (upper traces) and with 1 µM TRAM-34 added to the pipette solution. Insets show the effects of 10 µM noradrenaline. (**B**) The peak IsAHP current, measured 500 ms after the voltage step is plotted under the two conditions in control and TRAM-34 loaded neurons at the indicated times after onset of the whole-cell recording configuration (break-in). The lower panel shows the same data normalized to the amplitude 2 minutes after break-in. (**C**) Current clamp recordings from the neurons shown in (**A**), discharge evoked by a 600 ms current injection. (**D**) Plotted are the action potential half width and number of evoked action potentials evoked by the current injection at the indicated times in control and TRAM-34.

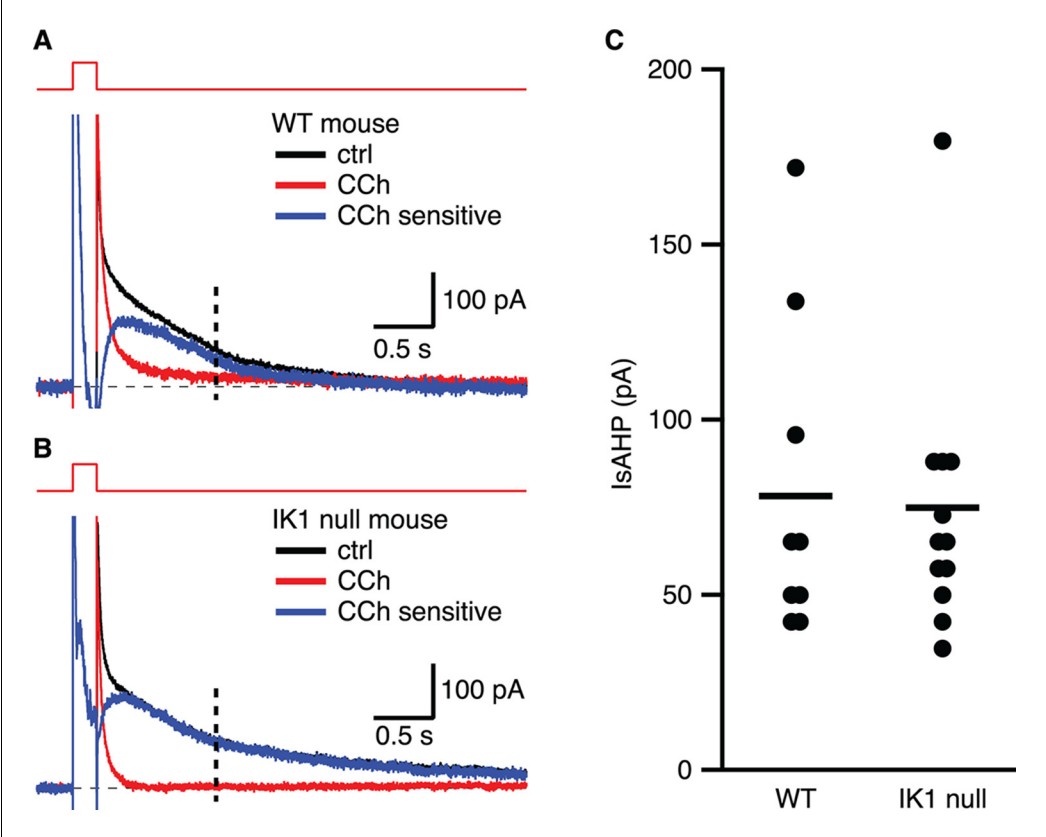

**Figure 7.** The IsAHP in IK1 null mice is not different from wild type. (A,B) Tail currents elicited by the voltage protocol from −63 to 7 mV shown above the traces for wild type (A) and IK1 null (B). Black traces are control, red traces are after CCh application, and blue traces are the subtracted CCh sensitive currents. (C) Scatter plot of the IsAHP amplitude for wild type and IK1 null mice. Mean indicated by horizontal line.

dependent, voltage-independent K$^+$-selective channels. Members of the SK (KCNN) channel family share many features that are similar to those that mediate the sAHP. Thus, all four members (KCNN1-4; SK1-3, IK1) are voltage independent and are Ca$^{2+}$-gated via constitutively bound cal-modulin (*Adelman et al., 2012*). The fourth member of the family, IK1, has a larger unitary conductance than the other family members (~40 pS vs ~10 pS in symmetrical K$^+$) (*Kohler et al., 1996*; *Ishii et al., 1997*; *Joiner et al., 1997*) and, importantly, the SK and IK channels are pharmacologically distinct. The peptide toxin, apamin potently and selectively blocks SK2 and SK3 channels (*Adelman et al., 2012*), while IK1 channels are apamin-insensitive but are selectively blocked by TRAM-34 (*Wulff et al., 2000*). IK1 is additionally blocked by the scorpion peptide charybdotoxin (ChTX) (*Ishii et al., 1997*) that also blocks BK channels. I*n situ* hybridization and immunohistochemistry data indicate that SK1-3 are expressed in overlapping but distinct patterns in the CNS. In hippocampal CA1 pyramidal neurons and BLA pyramidal neurons, SK2 is heavily expressed while SK1 and SK3 are expressed at lower levels (*Stocker and Pedarzani, 2000*; *Sailer et al., 2002*). IK1 mRNA is expressed in peripheral tissues such as smooth muscle endothelium, gastrointestinal tract, lung, and salivary glands, but brain expression is limited (*Begenisich et al., 2004*).

Indeed, recent transcriptome profiling using single CA1 hippocampal pyramidal neurons did not detect significant IK1 expression (*Zeisel et al., 2015*). In contrast, a recent study using both immunohistochemistry and a GFP reporter mouse suggested that IK1 expression was significant in both cortex and hippocampus, including CA1 pyramidal neurons (*Turner et al., 2015*). Using a monoclonal antibody to IK1, immunohistochemistry detected IK1 throughout the hippocampal formation (*Turner et al., 2015*). However this antibody also recognized a band on Western blots using tissue derived from either of two independently generated IK1 knockout mice (*Turner et al., 2015*).

Therefore, the specificity of the monoclonal antibody is questionable. For both of these transgenic IK1 gene disruption mouse lines previous work reported the loss IK1 mRNA expression (*Si, 2006*; *Begenisich et al., 2004*), and for one line, the line employed here, the loss of IK1 protein expression as well (*Si, 2006*). Indeed, *Si et al. (2006)* used a different IK1 antibody that detected robust expression in Western blots with red blood cells prepared from wild type but did not detect a band in protein samples from the null mice, engendering confidence in this IK1 null mouse. We used these IK1 null mice to record the characteristic IsAHP in CA1 pyramidal neurons. Additionally, *Turner et al. (2015)* generated an IK1 promoter GFP-reporter mouse. In this mouse GFP was detected throughout the hippocampus. Assuming that the BAC employed contained all of the requisite regulatory sequences to direct normal IK1 expression, these results suggest that the IK1 promoter is active in hippocampus, but stands in contrast to transcriptome results (*Zeisel et al., 2015*). In red blood cells, IK1 has been shown to be the 'Gardos' channel, responsible for $Ca^{2+}$-dependent $K^+$ efflux. The channel protein is presumably very stable as erythrocytes live for ~120 days after enucleation. Thus it is possible that a very brief 'pulse' of IK1 mRNA could give rise to channels that are stable long after the mRNA has been degraded.

CA1 pyramidal neurons from SK1, SK2, or SK3 knockout mice do not show alterations in the IsAHP, suggesting that none of these channels individually contribute to the sAHP, particularly SK1 that is less apamin sensitive (*Bond, 2004*). In primary afferent neurons of the gastrointestinal tract there is a late AHP that follows the action potential that bears many similarities to the sAHP in CA1 pyramidal neurons. IK1 is prominently expressed in these myenteric neurons and the late AHP is blocked by TRAM-34 (*Nguyen et al., 2007*). Thus, it is likely that IK1 channel activity is responsible for this late AHP. A recent report showed that the sAHP in CA1 pyramidal neurons was at least partially blocked by the signature agent, TRAM-34, and that the sAHP was absent in IK1 null mice. The authors concluded that IK1 channel activity is also responsible for the sAHP in CA1 pyramidal neurons (*King et al., 2015*). Consistent with this, PKA has been shown to inhibit IK1 channel activity via direct phosphorylation of the channel (*Wong and Schlichter, 2014*).

We re-examined the sAHP in hippocampal CA1 pyramidal neurons and found that TRAM-34 or ChTX did not block it, and TRAM-34 had no effect on intrinsic excitability. Similar results were obtained from pyramidal neurons of the BLA. We noted that the average amplitude of the IsAHP in *Figure 1* is higher than previously recorded in CA1 pyramidal cells (*Gu et al., 2005*). There are two aspects to this. First, the IsAHP increases with age, and for the IsAHP in CA1 pyramidal neurons we used older (8 weeks) rats. Second, the amplitude of the IsAHP is sensitive to changes in temperature (*Lancaster and Adams, 1986*; *Sah and Isaacson, 1995*) that may explain the larger IsAHP amplitude in our present study (recorded at 22–23°). However, this is unlikely to affect our overall interpretation, since IsAHP may be increased at lower temperatures and therefore any IK1 component might be even bigger. Importantly, IK1 null mice showed a prominent IsAHP. Moreover, IK1 channels have been reported to have a single channel conductance of ~12 pS (*Joiner et al., 1997*), however, while not directly recorded, channels underlying the IsAHP have been proposed to have a much lower single channel conductance of 2– 5 pS (*Sah and Isaacson, 1995*). Our data show no effect of TRAM-34 in contrast to *King et al. (2015)* This discrepancy may be related to a combination of several factors: 1) run-down of IsAHP during whole-cell recordings has been observed in several previous studies and cannot be excluded in absence of time course plots. 2) The $Ca^{2+}$ influx and IsAHP may differ during synaptic stimulation (*King et al., 2015*) or somatic current pulses by recruitment of different $Ca^{2+}$ sources. This may affect activation of IK1 depending on its specific subcellular localization. Regardless, data from three different laboratories using different conditions failed to find an effect of TRAM-34. Therefore, we conclude that the channel(s) underlying the sAHP is not IK1 and has not yet been identified. Several possibilities remain. For example, the sAHP might not be due to the activity of one particular type of K channel, but rather an ensemble of channels (*Andrade et al., 2012*). It has also been proposed that the sAHP is mediated at least in part by Kv7/KCNQ (M) channels with $Ca^{2+}$ sensitivity endowed by the calcium binding protein, hippocalcin (*Tzingounis et al., 2010*; *Tzingounis et al., 2007*), although the Kv7/KCNQ (M) channel blocker XE991 fails to block the sAHP of CA1 pyramidal cells (*Gu et al., 2005*) (see *Figure 4*). Alternatively, the $K^+$ channel superfamily contains several subunits that do not express functional channels on their own. In either case, the pore-forming subunits may be co-assembled into larger signaling complexes that endow $Ca^{2+}$ sensitivity and characteristic slow activation kinetics. Recapitulating the sAHP using cloned

components, together with gene targeting and pharmacology will eventually reveal the molecular details of the sAHP.

## Materials and methods

### Animal handling and slice preparation

All procedures were done in accordance with the guidelines of the Institutional Animal Care and Use Committee (IACUC) of the Oregon Health & Science University (IACUC: IS00002421), the Animal Care and Use Committee of Institute of Basic Medical Sciences of the University of Oslo (FOTS ID 5676), and the Animal Ethics Committee (AEC) of the University of Queensland (QBI/551/12/ NHMRC/ARC). Acute hippocampal slices were prepared from 3–8 week-old Wistar rats, IK1 null mice (*Si, 2006*) or C57BL/6J mice. Acute amygdala slices were prepared from 3–4 week-old C57BL/ 6J mice. Slices from rats and mice were prepared as previously described (*Gu et al., 2005*; *Lin et al., 2008*; *Faber et al., 2005*).

Organotypic hippocampal slice cultures were prepared from postnatal day 5–6 Wistar rats and used for recordings 14–21 d after preparation. Pups were decapitated and the brains placed in a solution containing (in mM): NaCl 137, KCl 5, $NaH_2PO_4$ 0.85, $CaCl_2$ 1.5, $KH_2PO_4$ 0.22, $MgSO_4$ 0.28, $MgCl_2$ 1, $NaHCO_3$ 2.74 and glucose 45, dissolved in tissue grade water. Each hippocampus was individually dissected out and cut into 400 µm thick transverse slices using a McIlwain tissue chopper. Slices were placed on filter membranes (0.4 um Hydrophilic PTFE filters, Millipore, Billerica, MA), and cultivated in 6-wells plates at 36 °C for 21 d. Each of the wells contained 1 ml culture medium replaced first after 24 h, and then every 3–4 days. The culture medium consisted of Basal Medium Eagle with HBSS (50%) and Hanks balanced salt solution, HBSS (25%) (both from AMIMED, UK), heat inactivated horse serum (25%), penicillin/streptomycin (100 U/ml) and (in mM): L-glutamine 1, glucose 20 and $NaHCO_3$ 6.

### Electrophysiology

For voltage clamp recordings, CA1 and BLA pyramidal neurons from acutely prepared brain slices were visualized with infrared–differential interference contrast optics (Zeiss Axioskop 2FS or Olympus BX50-WI) and a CCD camera (Sony, Tokyo, Japan or Dage-MTI, Michigan City, IN). Whole-cell patch-clamp recordings were obtained from CA1 pyramidal cells using an Axopatch 1D amplifier (Molecular Devices, Sunnyvale, CA), digitized with an ITC-16 analog-to-digital converter interface (Heka Instruments, Bellmore, NY) and transferred to computer using Patchmaster (Heka Instruments, Bellmore, NY) or a Multiclamp 700B amplifier (Molecular Devices, Sunnyvale, CA), digitized using a Digidata 1440A interface (Molecular Devices, Sunnyvale, CA), and transferred to a computer using pClamp10 software (Molecular Devices, Sunnyvale, CA). Recordings were performed at 22– 23°C. For CA1 pyramidal neurons, apamin (100 nM) was added to minimize contribution of SK currents, and SR95531 (2 µM) and CGP55845 (1 µM) were present to block $GABA_A$ and $GABA_B$ receptors, respectively. For BLA pyramidal neurons whole-cell patch-clamp recordings were obtained using a Multiclamp 700A amplifier (Molecular Devices, Sunnyvale, CA, USA), digitized with an ITC-16 analog-to-digital converter interface (Heka Instruments, Bellmore, NY) and transferred to computer using Axograph-X (Axograph Scientific, New South Wales, Australia). Recordings were performed at 32°C.

Patch pipettes (2.5–3.5 MΩ) for IsAHP recordings in CA1 pyramidal neurons were filled with a $KMeSO_4$ internal solution containing (in mM) $KMeSO_4$ 140, NaCl 8, $MgCl_2$ 1, HEPES 10, MgATP 5, $Na_3GTP$ 0.4, EGTA 0.05, (pH 7.3). For IsAHP recordings in BLA neurons, patch pipettes (4–6 MΩ) were filled with a $KMeSO_4$ internal solution containing (in mM) $KMeSO_4$ 135, NaCl 8, HEPES 10, $Mg_2ATP$ 2, $Na_3GTP$ 0.4, Spermine 0.1, Phosphocreatine 7, EGTA 0.2, (pH 7.3 with KOH; osmolarity ~ 290). Currents were recorded in whole-cell voltage clamp mode. For CA1 and BLA pyramidal neurons the membrane potential was held at −63 mV (CA1) and −50 mV (BLA), and IAHP currents were evoked in CA1 pyramidal neurons by depolarizing voltage commands to +7 mV for CA1 neurons for 200 ms and 10 mV for BLA neurons for 100 ms followed by a return to baseline potential for 10 sec where the current underlying the sAHP was measured. All cells had a resting membrane potential more hyperpolarized than −60 mV and input resistances of 150–350 MΩ for CA1 pyramidal neurons and 75–325 MΩ for BLA pyramidal neurons. Input resistance was determined from a −5 mV (100 ms)

hyperpolarizing pulse applied at the beginning of each sweep. Access resistance was 80% electronically compensated and stable at <20 MΩ. IAHP recordings were filtered at 3 kHz and digitized at a sampling frequency of 10 kHz. Voltages in CA1 and BLA pyramidal neurons were corrected for the liquid junction potential. Data were analyzed using Igor Pro (WaveMetrics, Lake Oswego, OR) or Excel (Microsoft, Seattle, WA). Data are expressed as mean ± SEM. Paired t-tests or Wilcoxon-Mann-Whitney 2-sample rank test was used to determine significance; $P < 0.05$ was considered significant.

For current clamp recordings, organotypic hippocampal slice cultures or acutely cut hippocampal slices were transferred to a recording chamber perfused with aCSF (34 °C) of the following composition (in mM): NaCl 125, KCl 3.5, $MgCl_2$ 1, $NaH_2PO_4$ 1.25, $NaHCO_3$ 25, $CaCl_2$ 1.6, glucose 25. Acutely cut BLA brain slices were transferred to a recording chamber perfused with aCSF (34°C) of the following composition (in mM): NaCl 118, KCl 2.5, $MgCl_2$ 1.3, $NaH_2PO_4$ 1.2, $NaHCO_3$ 25, $CaCl_2$ 2.5, glucose 10. CA1 and BLA pyramidal neurons were visually identified using infrared-differential interference contrast (IR-DIC) optics on an Olympus BX-51WI or BX-50WI microscope. For hippocampal neurons the intracellular recording solution contained (in mM): KGluconate 120, KCl 20, $Na_2$. phosphocreatine 5, HEPES 10, MgATP 4, $Na_2$GTP 0.4, EGTA 0.1 (pH 7.2 adjusted with KOH). For BLA neurons the intracellular recording solution contained in (in mM) $KMeSO_4$ 135, NaCl 8, HEPES 10, $Mg_2$ATP 2, $Na_3$GTP 0.4, Spermine 0.1, Phosphocreatine 7, EGTA 0.2, (pH 7.3 with KOH). Whole-cell current clamp recordings were obtained using a Multiclamp 700A or B amplifier, signals were low-pass filtered at 10 KHz and digitized at 20 KHz. Access resistances, typically 15–35 MΩ, were monitored and compensated throughout the experiments. Potentials were corrected for liquid junction potential (−14 mV). Origin 9.1 (Hearne Scientific Software, Victoria, AU) was used for statistical analysis and graphical representations.

In all hippocampal experiments, 6,7-dinitroquinoxaline-2,3-dione (DNQX, 10 µM), DL-2-amino-5-phosphono-pentanoic acid (DL-AP5, 50 µM) and gabazine (SR-95531, 5 µM) were added to the aCSF to block spontaneous synaptic transmission. In *Figure 4*, XE991 (10 µM), apamin (100 nM) and TRAM-34 (1 µM) were added to the ACSF and slices were perfused for 25 minutes before recordings started. The mAHP and sAHP values were measured by averaging a time window (20 and 100 ms, respectively) around the peak of each one (20–50 ms and 0.17–0.4 s after a train of 7–8 spikes, respectively) and subtracted from the baseline voltage. Statistical analysis was done in Minitab 17 (Minitab UK). Group data are expressed as mean and standard error (SEM, in parentheses), and the sample size of cells (*n*), and number of rats (N) used in the individual experiments is given. Data were checked for normal distribution according to a normal probability plot of their residuals and 2-tailed *t* tests were performed for independent samples (control group vs. drug group) using a critical level of significance α=0.05.

HEK293 cells (ATCC CRL-1573) were transfected using lipofectamine with a plasmid directing expression of IK1 cDNA (*Ishii et al., 1997*).

## Acknowledgements
We thank Drs. Mark Taylor and Mike Lin for the IK1 null mice. We thank Ms. Cecilie Petterson Oksvold for preparing organotypic hippocampal slice cultures.

## Additional information
### Funding

| Funder | Grant reference number | Author |
| --- | --- | --- |
| National Institutes of Health | NS03888 | Jim Maylie<br>John P Adelman |
| National Institutes of Health | MH093599 | Jim Maylie<br>John P Adelman |
| Norges Forskningsråd | | Johan F Storm |
| Australian Research Council | SR120300015 | Pankaj Sah |

The funders had no role in study design, data collection and interpretation, or the decision to submit the work for publication.

## Author contributions

KW, PMA, CH, VR, WWW, MCR, PS, JM, JFS, Conception and design, Acquisition of data, Analysis and interpretation of data, Drafting or revising the article; JPA, Conception and design, Analysis and interpretation of data, Drafting or revising the article

## Ethics

Animal experimentation: All procedures for this study were done in accordance with the guidelines of and were approved by the Institutional Animal Care and Use Committee (IACUC) of the Oregon Health & Science University (IACUC: IS00002421 for rats and IK1 null and wild type mice), the Animal Care and Use Committee of Institute of Basic Medical Sciences of the University of Oslo (FOTS ID 5676 for rats), and the Animal Ethics Committee (AEC) of the University of Queensland (QBI/551/12/ NHMRC/ARC for rats). All surgery was performed under isoflurane anesthesia, and every effort was made to minimize suffering.

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
