## [Decision Letter]

Thank you for submitting your work entitled "IK1 Channels Do Not Contribute to the Slow Afterhyperpolarization in Pyramidal Neurons" for consideration by *eLife*. Your article has been favorably evaluated by Richard Aldrich (Senior editor) and three reviewers, one of whom, Sacha Nelson, is a member of our Board of Reviewing Editors.

The following individual involved in review of your submission has agreed to reveal their identity: Robert Foering (peer reviewer).

The reviewers have discussed the reviews with one another and the Reviewing editor has drafted this decision to help you prepare a revised submission.

Summary:

In this manuscript, three independent labs have pooled resources to test whether IK1 channels mediate the sAHP and IsAHP in pyramidal neurons from CA1, as reported by King et al. (2015). In addition, the authors examined AHP currents in an IK1 KO animal and examined IsAHP and the sAHP in pyramidal cells from the basolateral amygdala. The principal findings are in direct opposition to those of King et al. (2015). The current study found that the IK1 KO had normal IsAHP currents and that the reported IK1-selective blocker TRAM-34 had no effect on the sAHP or IsAHP in either CA1 (acute or organotypic slices) or basolateral amygdalar pyramidal neurons (despite clear effects of TRAM-34 on IKI currents in HEK293 cells).

The data shown are convincing and strongly support the conclusion that IK1 is not a major player in the sAHP in CA1 or amygdalar pyramidal cells. Although these are essentially negative results, the present manuscript provides an important caution for interpretation of the role of IK1 in generating the sAHP.

Essential revisions:

1) One reviewer was concerned that there are inconsistencies in the treatment and reporting of data between figures, likely reflecting the fact that they were generated separately by independent laboratories..

2) In addition, there was concern that the all of the TRAM-34 data in this paper and the paper it fails to reproduce used a single dose. Given the different preparation employed and the striking difference in results compared with King et al., it would seem that at least some of the CA1 data should be pursued with other doses to ensure that the differences do not simply reflect different efficacy of the drug between labs.

3) The current values reported in Figure 1 seem to be much bigger than the values reported by other studies in CA1 pyramidal neurons (Gu et al., 2008; Pedarzani et al., 1998). Is it possible that the conditions used in these experiments increased the activity of Ca^2^ -activated K channels; or that additional currents are isolated? Could this affect the overall interpretation? This does not necessarily require any additional experiments, but should be addressed in the text.

---

## [Author Response]

*Essential revisions:*

*1) One reviewer was concerned that there are inconsistencies in the treatment and reporting of data between figures, likely reflecting the fact that they were generated separately by independent laboratories..*

We appreciate the reviewer’s concern and have done our best to address this, as detailed below. We believe that one of the strengths of the paper is that three groups, each with an established history of examining the sAHP, worked independently, each using slightly different techniques and measures, and came to the same conclusion.

*2) In addition, there was concern that the all of the TRAM-34 data in this paper and the paper it fails to reproduce used a single dose. Given the different preparation employed and the striking difference in results compared with King et al., it would seem that at least some of the CA1 data should be pursued with other doses to ensure that the differences do not simply reflect different efficacy of the drug between labs.*

We have repeated measuring the IsAHP in CA1 pyramidal neurons using 5 μM TRAM-34. The result is the same, no effect on the IsAHP. These data are now included in the text.

*3) The current values reported in Figure 1 seem to be much bigger than the values reported by other studies in CA1 pyramidal neurons (Gu et al., 2008; Pedarzani et al., 1998). Is it possible that the conditions used in these experiments increased the activity of Ca^2^ -activated K channels; or that additional currents are isolated? Could this affect the overall interpretation? This does not necessarily require any additional experiments, but should be addressed in the text.*

There are several aspects to this, and we assume the reviewer refers to Gu et al., 2005 (no voltage clamp data in Gu et al., 2008). First, the IsAHP increases with age and for our experiments on the IsAHP in CA1 pyramidal neurons we used older rats (8 weeks). A second point is temperature. The IsAHP tends to be larger at lower temperatures, and in Gu et al., 2005 the IsAHP was recorded at 30 °C and here, it was recorded at 22-23 °C. Pedarzani et al., 1998 recorded at room temperature as well (21-24 °C) but the internal solution was K-gluconate, which explains the smaller current amplitude (see also Kaczorowski, Disterhoft and Spruston, 2007). On one hand, the referee has a point, since King et al., 2015 recorded at 32-34 ºC and in our manuscript the IsAHP experiments in CA1 were done at 22-23 ºC while the rest of experiments were done at 34 ºC. On the other hand, this is unlikely to affect the overall interpretation, since Ca^2^ -dependent K currents may be increased at lower temperatures (and therefore any IK1 component might be even bigger). Therefore, we believe the fact that we showed no effect, even at different temperatures actually strengthens our conclusions. We have added this to the Discussion.